# Co-Overexpression of TWIST1-CSF1 Is a Common Event in Metastatic Oral Cancer and Drives Biologically Aggressive Phenotype

**DOI:** 10.3390/cancers13010153

**Published:** 2021-01-05

**Authors:** Sabrina Daniela da Silva, Fabio Albuquerque Marchi, Jie Su, Long Yang, Ludmila Valverde, Jessica Hier, Krikor Bijian, Michael Hier, Alex Mlynarek, Luiz Paulo Kowalski, Moulay A. Alaoui-Jamali

**Affiliations:** 1Department of Otolaryngology Head and Neck Surgery, Sir Mortimer B. Davis-Jewish General Hospital, Montreal, QC H3T 1E2, Canada; ludmilafvalverde@gmail.com (L.V.); jessica.hrhier@gmail.com (J.H.); mhier@jgh.mcgill.ca (M.H.); alex.mlynarek@mcgill.ca (A.M.); 2Segal Cancer Centre and Lady Davis Institute for Medical Research, Sir Mortimer B. Davis-Jewish General Hospital, Departments of Medicine, Oncology, and Pharmacology and Therapeutics, Faculty of Medicine, McGill University, Montreal, QC H3T 1E2, Canada; jie.su2@mail.mcgill.ca (J.S.); krikor.bijian@mail.mcgill.ca (K.B.); 3Department of Head and Neck Surgery and Otorhinolaryngology, AC Camargo Cancer Center and National Institute of Science and Technology on Oncogenomics (INCITO), São Paulo 01509-010, Brazil; biomarchi@gmail.com (F.A.M.); lp_kowalski@uol.com.br (L.P.K.); 4School of Integrative Medicine, Tianjin University of Traditional Chinese Medicine, Tianjin 301617, China; longyangrich@hotmail.com

**Keywords:** oral squamous cell carcinoma, tumor invasion, metastasis, genomic, tumor-associated macrophages, EMT, TWIST1, CSF1

## Abstract

**Simple Summary:**

There is clinical evidence that ulcerated and inflammatory cell-infiltrated oral cancer is frequently associated with early metastases. Our results from genomic screening in patients with metastatic oral cancer identified specific changes in genes that regulate macrophage chemotaxis and drive tumor progression. This opens up potential therapeutic opportunities toward personalized medicine tailored to manage patients with advanced disease.

**Abstract:**

Invasive oral squamous cell carcinoma (OSCC) is often ulcerated and heavily infiltrated by pro-inflammatory cells. We conducted a genome-wide profiling of tissues from OSCC patients (early versus advanced stages) with 10 years follow-up. Co-amplification and co-overexpression of *TWIST1*, a transcriptional activator of epithelial-mesenchymal-transition (EMT), and colony-stimulating factor-1 (*CSF1*), a major chemotactic agent for tumor-associated macrophages (TAMs), were observed in metastatic OSCC cases. The overexpression of these markers strongly predicted poor patient survival (log-rank test, *p* = 0.0035 and *p* = 0.0219). Protein analysis confirmed the enhanced expression of TWIST1 and CSF1 in metastatic tissues. In preclinical models using OSCC cell lines, macrophages, and an in vivo matrigel plug assay, we demonstrated that *TWIST1* gene overexpression induces the activation of *CSF1* while *TWIST1* gene silencing down-regulates *CSF1* preventing OSCC invasion. Furthermore, excessive macrophage activation and polarization was observed in co-culture system involving OSCC cells overexpressing *TWIST1*. In summary, this study provides insight into the cooperation between *TWIST1* transcription factor and *CSF1* to promote OSCC invasiveness and opens up the potential therapeutic utility of currently developed antibodies and small molecules targeting cancer-associated macrophages.

## 1. Introduction

Metastasis development and high rates of tumor recurrence are common events seen in patients with oral cancer and continue to pose a major clinical challenge [1,2,3]. Oral cancer is the most common subtype of malignant tumors of the head and neck, which represents the 6th most frequent cancer worldwide with approximately half million cases and 300,000 deaths annually [4,5,6]. Advances in surgical procedures and therapeutic approaches for oral squamous cell carcinoma (OSCC) have led to a substantial improvement in survival rates but the overall five-year relative survival is lower than 50% and it remains inferior compared to highly frequent cancers such as breast, prostate and lungs cancers [7,8]. The high incidence of tumor relapse and distant metastasis are the main contributors of OSCC-related mortality [9].

OSCC progression to metastasis involves a complex and partially understood interplay of factors involving tumor cells, their microenvironment, and the host. In particular, signaling pathways that regulate cell plasticity and heterotypic tumor-inflammatory cell interactions are critically determinant for OSCC progression [10]. Enhanced tumor cell plasticity is contributed primarily through the epithelial-mesenchymal-transition (EMT) process [11], a mechanism mediated by a network of transcriptional regulators, including the basic helix-loop-helix (bHLH) transcriptional factor TWIST1, to promote cell migration and metastasis development [12]. Furthermore, clinical and experimental evidence support that high levels of infiltrating tumor-associated macrophages (TAMs) generally promote cancer progression [13] and can predict poor patient prognosis [14,15]. Colony-stimulating factor-1 (CSF1), also known as macrophage colony-stimulating factor, is the major chemotactic agent for TAMs that controls the production, recruitment, and survival of macrophages in tumor microenvironment [16]. The effects of CSF1 are mediated by the CSF1 receptor tyrosine kinase (CSF1R), which upon autophosphorylation of CSF1R can trigger downstream signaling cascades implicated in transcriptional and translational regulation of genes involved in cell cytoskeletal remodeling, survival, proliferation, and differentiation [17]. Activation of CSF1R by its ligand has been shown to regulate invasiveness and anchorage-independent growth in cancer cells [18]. Through post-translational modification and alternative splicing, CSF1 can either be secreted into the blood stream as a glycoprotein, or chondroitin sulfate-containing proteoglycan, or expressed as a membrane-spanning glycoprotein on the surface of synthesizing cells [19,20]. Both cell-surface isoforms and secreted CSF1 can have a broad implication in the regulation of tumor-associated inflammatory responses [21]. In this study, we conducted a comprehensive genome-wide screening, and it was identified a common co-amplification of *TWIST1-CSF1* in cancer tissues using a unique cohort of patients with highly metastatic OSCC compared with non-metastatic ones. The clinicopathological impact of this co-overexpression was validated in a large cohort of OSCC patients. We further investigated the mechanistic implication of *TWIST1-CSF1* signaling to macrophage chemotaxis and polarization by showing a role of TWIST1 in the remodeling of OSCC tumor microenvironment via CSF1 regulation; this enhanced OSCC progression and metastasis competence in vitro and in vivo.

## 2. Results

### 2.1. TWIST1 and CSF1 Are Clustered in the Gained/Amplified Chromosomal Regions in Patients with Highly Metastatic Oral Cancer

Genomic analysis combined with clinicopathological data from our cohort containing non-metastatic OSCC (*n* = 10) vs. advanced and highly metastatic OSCC patients (*n* = 10) with 157 months follow-up, identified a set of genes that are selectively overexpressed in metastatic compared to non-metastatic OSCCs (Appendix A). Regardless of the clinical features, the unsupervised hierarchical analysis of the aGCH data allowed the clustering of OSCC cases, indicating that these markers are potentially relevant to oral tumor progression. Specifically, the gene expression levels of *CDH1*, *SNAIL*, *TGFBR1*, *ZEB2*, *PAX3*, *FOXO1*, *RB1*, *MYCN*, *NOG*, *LEF1*, *FGFR2*, *ESRP1*, *FBLN5*, *TNF*, *FGF1*, *AKT2*, *CDH11*, *SOX10*, *SMAD3*, *NFYB*, *NOTCH2*, *HEY1*, *NDRG1*, *MST1R*, *JAG1*, *AKT1*, *GLIS2*, *AGT*, *CSF1*, *TWIST1*, *PTPN14*, *SMURF2*, *TRIM28*, *CSFR1*, *LOXL2*, *STMN1*, *KLF8*, *WNT2*, *FOXS1*, *HAS2*, and *PDPK1* were able to dissociate into two different groups of advanced/metastatic and non-metastatic patients (Figure 1A). The prevalence of common genomic amplifications was seen in chromosomal site and highest scoring locus identified in this analysis were mapped to regions of recurrent copy number gain in metastatic oral cancer, including 1p13.3 and 7p13, which correspond to *CSF1* and *TWIST1* (Figure 1B). The plots showing the frequency of copy number losses (red) and gains (blue) comparing early/moderate stage (non-metastatic) vs. aggressive/metastatic OSCC identified chromosomal imbalances in more than 90% of invasive cases related to *CSF1* and *TWIST1* (Figure 1C).

### 2.2. TWIST1 and CSF1 Can Predict Oral Cancer Tumor Progression and Poor Outcomes

TWIST1, CSF1, and CD68 proteins expression levels were investigated in relation to the clinicopathological parameters using an independent cohort of tissue samples from 141 patients with oral cancer (Appendix A) who had tumor relapse (*n* = 44; 31.2%) or distant metastasis and patients ith good outcomes (*n* = 97; 68.8%). In agreement with the genomic analysis, overexpression of TWIST1 and CSF1 protein levels was seen in advanced stages. A weak staining was observed in morphologically normal tissues while a strong nuclear staining (for TWIST1) and cytoplasmic staining (for CSF1 and CD68) was detected in OSCC samples (Figure 2A). High expression levels of TWIST1, CSF1, and CD68 proteins were observed in patients with oral cancer at advanced stages (total expression values of TWIST1, CSF1, and CD68 were 63.5%, 75%, and 60.8%, respectively), while in normal oral tissues surrounding OSCC, the expression was very weak to undetectable (Figure 2B). The survival probability shows a mean five-year overall survival rate of 35.2 months (Kaplan-Meier method, period of 1 to 120 months; SD ± 9). To investigate whether TWIST1, CSF1, and CD68 expression was associated with patients’ outcomes, Kaplan–Meir and Cox proportional hazard models were performed. A worst overall survival probability was experienced by patients with TWIST1 (log-rank test, *p* = 0.0035; adjusted HR 7.837 (95% CI: 1.099–50.880; *p* = 0.040) and CSF1 overexpression (log-rank test, *p* = 0.0219; adjusted HR 2.182 (95% CI: 0.993–4.796; *p* = 0.052)) but not CD68 (log-rank test, *p* = 0.3390; adjusted HR 1.004 (95% CI: 0.987–1.021; *p* = 0.628)) (Figure 2B).

### 2.3. TWIST1 Regulates CSF1 Expression in OSCC Cells

A panel of oral cancer human cell lines (SCC9, SCC25, and OSCC1.2/RBT3) where *TWIST1* is overexpressed or knockdown, as well as normal immortalized oral epithelial cells (NOE) established from human tongue, were investigated as a preclinical model. NOE cells undergo EMT upon stimulation by TGFβ (Figure 3A). In comparison to the NOE cells, relative quantification of transcript levels of *TWIST1* and *CSF1* revealed an elevated expression in OSCC cells (SCC9 and SCC25), and highest expression in the poorly differentiated and metastatic cell line (OSCC1.2) established from an advanced human OSCC (stage: T4N2b) with vascular, lymphatic and perineural invasion [18] (Figure 3B, *p*  <  0.05). Further investigation showed significant down-regulation of *CSF1* in OSCC cells after *TWIST1* gene silencing, but the gene expression was increased after the treatment with TGFβ (Figure 3C). The amounts of secreted and total CSF1 protein concentrations were measured in cells where *TWIST1* is induced or inhibited using ELISA and immunoblotting, respectively. The results showed low protein concentrations of CSF1 in cells where *TWIST1* was downregulated and high CSF1 protein levels in cells where *TWIST1* expression was induced following exposure to TGFβ (Figure 3D,E). Furthermore, we investigated in silico whether CSF1 genomic regions show a high degree of similarity with genes directly regulated by *TWIST1* (e.g., CDH11, RAB39B, GADD45A, SEMA3C). Specifically, we compared the occurrence of E-box responsive elements 5′-CANNTG-3 targeted by TWIST1 through multiple alignments with genomic region (+/−5000 bp) using BL2SEQ BLAST (http://blast.ncbi.nlm.nih.gov). Then, the identified region was amplified by PCR and cloned into pGL3 luciferase reporter plasmid to test its capabilities in driving luciferase expression in cells that have detectable basal levels of *TWIST1*. Transient transfection of pMCSF-R(m40)-luc (5′-ATCGGTACC-3′) into OSCC cells with or without TGFβ treatment revealed that, although the mutated constructs displayed constitutive baseline activity, this was increased after TGFβ treatment (Figure 3E,F).

### 2.4. TWIST1 Regulates Macrophage Polarization and Chemotaxis during OSCC Progression

The relevance of *TWIST1* regulation to TAM chemotaxis was evaluated using conditioned medium from OSCC cells where *TWIST1* is overexpressed or down regulated. Macrophage recruitment was evaluated in a co-culture system using modified Boyden chamber assay where heterotypic interaction between macrophage and OSCC cells was investigated. RAW264.7 were co-cultured with AT84 cells, AT84 expressing scrambled sh*TWIST1* or AT84 sh*TWIST1*. In addition, we also investigated AT84 pBabe (control), AT84 overexpressing *TWIST1*, and their respective conditioned media. The results reveal that *TWIST1* inhibition leads to decreased macrophage recruitment to AT84 cancer cells (Figure 4A). The phenotypic differences between two functional states of macrophage polarization (M1 and M2) were evaluated (Figure 4B) and indicated changes towards M2 phenotype, which were further confirmed through the expression of selected markers of macrophage polarization (Nos2, Tnf, Il6, Il12, Chil3, Retnla, Arg1, Il10, Nr3c2). It was observed down-expression of classically activated M1 macrophage genes (Nos2, Tnf, Il6, Il12) followed by increased mRNA expression of polarized M2 genes (Chil3, Retnla, Arg1, Il10, Nr3c2) (Figure 4C). Polarized M2 macrophages were predominant in the OSCC cell overexpressing *TWIST1*.

To further confirm the impact of *TWIST1* on the host microenvironment in relation to macrophage infiltration, in vivo Matrigel plug assay was used (Figure 5A). Matched *TWIST1*-proficient versus *TWIST1*-deficient cells embedded into Matrigel plugs were injected subcutaneously to mice and the resulting plugs were harvested 14 days later and processed for IHC staining to analyze the degree of cancer cell infiltration and TWIST1 and CSF1 protein expression. Negative staining for TWIST1 and a week expression for CSF1 was observed in sections from mice where AT84 cells sh*TWIST1* were implanted. In contrast, a strong TWIST1 nuclear staining and CSF1 cytoplasmic expression was detected in the plug with AT84 cells (Figure 5B). Equally important, *TWIST1* down-regulation inhibited tumor growth when AT84 cells were implanted either orthotopically into the tongues of mice and no clinical lesion could be observed (compared with AT84 control) (Figure 5C). No macroscopic lung lesions (metastasis) were macroscopically detected in *TWIST1*-deficient cells, whereas the control mice had to be killed because of the size of their lung lesions and the clinical signs of respiratory distress.

### 2.5. TWIST1 Reconstitution Increases Metastatic Potential

To investigate the functional role of *TWIST1* in OSCC, we studied this gene in a syngeneic model using the murine oral carcinoma cell line (AT84), which expresses high levels of *TWIST1* and also a highly metastatic human cell line (OSCC1.2/RBT3) using nude mice (Figure 6A). We knockout *TWIST1* expression in tumor cells using the CRISPR/Cas9 and the success of the gene inactivation was confirmed by more than 95% at both protein and mRNA levels (Figure 6A). There was a significant down-regulation of CSF1 at protein and gene expression level when *TWIST1* was knockout in AT84 and RBT3 (Figure 6A). However, protein and gene expression of both (*TWIST1* and *CSF1*) are present after *TWIST1* is reconstituted in the *TWIST1* knockout tumor cells (Figure 6A). We then compared tumor volume and weight features of this *TWIST1*-CRISPR1 selected clone with the control cells, and it was found a similar result for AT84 and RBT3 (Figure 6A). These In vivo studies showed a significant reduction (80%) in lung metastatic potential when *TWIST1* was knockout (Figure 6B,C). When *TWIST1* was reconstituted, the metastatic islands in the lung were observed macro and microscopically (Figure 6). These data suggest that the absence of *TWIST1* endows OSCC cells with a decreased invasive capacity in vivo, reducing the metastatic competence.

## 3. Discussion

Tumor–host and cell–cell interactions are critical for the remodeling of oral cancer microenvironment and susceptibility to progress to metastasis. In particular, dynamic interaction between OSCC cells and a multitude of stromal cells within tumor microenvironment, including fibroblast, monocytes, endothelial and immune cells, can dictate the risk of OSCC invasion. Among infiltrating host cells, activated TAMs and cytotoxic T lymphocytes (CTLs) represent major components of tumor-infiltrating cells [22]. Tumors with marked leukocyte infiltration are often associated with aggressive behavior and poor prognosis [23,24,25]. The molecular basis of these associations is not fully understood but enhanced CTLs and TAMs can suppress T cell activation and promote evasion of immune surveillance mechanisms to facilitate metastatic spread [26,27,28].

Here, we conducted genome-wide screening on tissue samples from non-metastatic vs. metastatic OSCC patients. The results validated in a panel of preclinical OSCC models as well as in a large cohort of OSCC patients showed significant changes in genes known to regulate EMT process. Seminal studies have established the EMT process to be orchestrated by complex networks involving signal transduction pathways (e.g., TGFβ, Wnt, and Notch) and transcriptional factors (e.g., Slug, Snail, Twist1, and Zeb1/2) [26,27]. In particular, *TWIST1* expression was found to be strongly elevated in patients that developed lung metastasis and was a predictor of the overall survival. *TWIST1*, a transcription factor that induces EMT by repressing E-cadherin, is organized as a highly conserved basic helix-loop-helix (bHLH) motif and a protein interacting region “Twist box” on the C-domain. Through the bHLH domain, *TWIST1* recognizes E-box responsive elements 5′-CANNTG-3′ and can behave as transcriptional repressor or activator, depending on the cellular context [29]. As well, *TWIST1* can form either homo- or heterodimers and cooperate with other nuclear factors such as Snail, MyoD, Runx, and MEF2 [30]. *TWIST1* is overexpressed in many advanced cancers and its expression levels have been correlated with poor outcomes [31,32,33]. Several growth factors and oncogenic proteins induce *TWIST1* transcription, including NF-κB signaling, inflammatory cytokines, *TMPRSS2/ERG* oncogenic fusion gene [34], and activated EGFR/ErbB2 receptors [35]. Although members of the TWIST family have been extensively investigated, the mechanisms by which *TWIST1* transcription factor regulates immune/pro-inflammatory response in the context of metastasis development remain partially understood.

We characterized the contribution of *TWIST1* as a potent inducer of OSCC-associated macrophages contributing to OSCC tissue remodeling and progression. In support of this, our profiling of progressive OSCC revealed a consistent overexpression of CSF1, a major chemotactic factor for TAMs. Interestingly, TAMs are able to inhibit (M1 type) or promote (M2 type) tumorigenesis [36]. *CSF1* has been implicated in the recruitment and polarization of M2, which once activated can release trophic cytokines and pro-angiogenic factors to enhance tumor cell growth [37,38,39,40]. M2 have limited tumor cell cytotoxicity and antigen-presenting capability but also can suppress lymphocytes activation [41]. Accordingly, the presence of specific TAM subtype infiltration in the tumor microenvironment ranges from either poor or favorable prognosis. *CSF1* overexpression is strongly associated with high incidence of recurrence and metastasis [42,43,44,45]. We observed that patients with advanced OSCC showed high expression of *TWIST1* and *CSF1*. Oral cancer cells undergoing to EMT may not only contribute to increase metastatic competence but may become resistant to cytotoxic T-lymphocytes. We also investigated the presence of T cells in epithelial tumor islets (intraepithelial tumor-infiltrating lymphocytes, TIL) by quantified all CD3^+^ T cells and cytotoxic CD8^+^ T cells (Appendix A). TIL status was differentially expressed comparing normal, OSCC and lymph nodes, but it was not statistically significant for OSCC patients’ outcomes. However, using the same cohort, *TWIST1* and *CSF1* were predictive of OSCC progression and poor prognosis. It should be noted that EMT is a reversible trans-differentiation program with inherent plasticity associated with the stemness of cancer cells sharing considerable redundancies such as mediators, factors, signal transducers and these are not induced simultaneously, We identified that the co-overexpression of TWIST1-CSF1 drives biologically aggressive phenotype in a pure epithelial cell population (our samples were microdissected) from patients with oral cancer presenting very similar clinicopathological characteristics and outcomes. However, to determine if patients may or may not respond to immunotherapy, future researchers should be able to measure the degree of tumor cell undergoing to EMT considering inter and intra-tumor heterogeneity associated with the microenvironment, which is heterogeneous as well.

Our preclinical studies showed that in the OSCC cells, the ectopic *TWIST1* overexpression increased *CSF1* expression while *TWIST1* gene silencing down-regulated *CSF1*. To clarify the diversity of macrophage recruitment within the tumor and their relevance for invasiveness, we confirmed the potency of OSCC-induced *TWIST1* expression to promote macrophage chemotaxis using the in vivo Matrigel plug assay, supporting that *TWIST1*-*CSF1* axis impacts on remodeling of OSCC microenvironment via recruitment and polarization of TAMs to promote pro-metastasis signaling. However, we cannot rule out alternative regulatory mechanisms for *TWIST1*. Indeed, we used computational methods in which we mined existing and predictive molecular interaction networks published in literature. We identified potential signaling patterns associated with tumor progression and metastatic competence in oral cancer by combining initial hub results from different centrality measures and randomly analysis for each network using the experimental results from our metastatic cohort (Figure 7). Protein enrichment involved with cancer stem cell signaling, EMT, and inflammatory process were identified as cores for cancer invasion and metastatic process related with *TWIST1* and *CSF1* overexpression. In summary, this study shows that the co-expression of *TWIST1*-*CSF1* is a common event in metastatic OSCC and drives a biologically aggressive oral cancer phenotype.

## 4. Materials and Methods

### 4.1. Study Population

Primary frozen samples from tongue of metastatic (*n* = 10) and non-metastatic patients (*n* = 10) followed-up for 157 months (over than 10 years) were surgically removed at the Department of Head and Neck Surgery (AC Camargo Cancer Hospital, Brazil). Two pathologists reviewed the slides to select appropriate areas for laser capture microdissection (LCM). The microdissected samples were used for genomics experiments (Appendix A). Technical validation (cohort from Brazil) and validation of the biological process (independent cohort from Canada) were done in 141 paraffin-embedded oral cancer specimens from patients who had tumor relapse (*n* = 44; 31.2%) or distant metastasis and patients with good outcomes (*n* = 97; 68.8%) were evaluated by immunohistochemistry (IHC) using tissue microarray (TMA). These patients were followed-up for 96.2 months. Tumor relapse was histologically confirmed, and patients were followed-up after treatment. Morphologically matched normal specimens from the surgical margins (clear of tumor cells) were included as controls (Appendix A).

Eligibility criteria included previously untreated OSCC patients submitted for treatment in the same institution without any distant metastasis at the diagnosis (M0). The tumor staging was re-classified according to the International Union Against Cancer (TNM) and grouped as early clinical stage (T1 + T2) or advanced clinical stage (T3 + T4) [46]. The medical records were the main source to obtain detailed clinicopathological information. Strengthening the reporting of observational studies (STROBE Statement) was used to ensure appropriate methodological quality (http://www.strobe-statement.org/).

### 4.2. Laser Capture Microdissection (LCM) and DNA Isolation

Approximately 3000 cells were captured from 5µm frozen tissue sections and genomic DNA was extracted using DNeasy Blood & Tissue Kit (Qiagen, Chatsworth, CA, USA) after LCM (PixCell^®^ II, Arcturus Engineering, Mountain View, CA, USA). Samples were evaluated with NanoDrop^®^ (Thermo Scientific, Wilmington, MA, USA) and Bioanalyzer (Agilent, Palo Alto, CA, USA) to determine the DNA concentration and quality.

### 4.3. Genome-Wide Screening and Analysis

OSCC samples (metastatic *versus* non-metastatic) and normal tissues (Promega, Madison, WI, USA) were differentially labeled using the Genomic DNA Enzymatic Labeling Kit (Agilent, Santa Clara, CA, USA). The hybridizations were performed on human aCGH 44K (Agilent) following the manufacturer’s recommendations. The image acquisition and statistical parameters to define alterations were as we previously described [46]. In brief, we considered an unsupervised clustering to identify the group profile considering copy number gain (≥0.6), copy number loss (≤−0.8), and homozygous loss (≤−1.2). Hierarchical cluster was done using Euclidean distance and average linkage with 1000 permutations.

### 4.4. Quantitative Real Time RT-PCR (qRT-PCR)

cDNAs were synthesized using Superscript II reverse transcriptase (Invitrogen, Carlsbad, CA, USA) and random primers (Invitrogen). Primer set sequences was design as following *TWIST1* (5′-TCCATTTTCTCCTTCTCTGGAA-3′; 5′-CCTTCTCGGTCTGGAGGAT-3), *CSF1* (5′-ATGACAGACAGGTGGAACTGCCAG-3′; 5′-TCACACAATTCAGTAGGTTCAGG-3). qRT-PCR amplification was conducted using Power SYBR Green^®^ Master Mix (Thermo Fisher, Carlsbad, CA, USA) and the quality controls steps followed MIQE Guidelines [19]. The reactions were done in triplicate. *GAPDH* was considered the stable control gene from four endogenous genes tested (*GAPDH* (5′-AATGAAGGGGTCATTGATGG-3′; 5′-AAGGTGAAGGTCGGAGTCAA-3), *ACTB* (5′-GCACCCAGCACAATGAAG-3′; 5′-CTTGCTGATCCACATCTGC-3), *HPRT1* (5′-GAACGTCTTGCTCGAGATGTGA-3′; 5′-TCCAGCAGGTCAGCAAAGAAT-3)*,* and *BCRP* (5′-CCTTCGACGTCAATAACAAGGAT-3′; 5′-CCTGCGATGGCGTTCAC-3) using the geNorm algorithm [47]. Relative gene expression analysis was done using Pfaffl model [48].

### 4.5. Preparation of the Tissue Microarray (TMA)

1.0 mm cores were extracted from previously microscopically defined OSCC representative areas and matched morphologically normal epithelium from adjacent margins free of tumor with a Tissue Microarrayer^®^ (Beecher Instruments, Silver Springs, MD, USA) [31,46]. Tissue cores were punched and arrayed in duplicate on a single recipient TMA paraffin block. Each core was spaced 0.2 mm apart. After cutting sections from the recipient block, the slides received a layer of paraffin to prevent oxidation and stored at −20 °C.

### 4.6. Immunohistochemistry (IHC) and Statistical Analysis

Immunohistochemistry reaction was carried out on the TMA as we described [31,46]. In brief, the slides were incubated with primary antibodies diluted in PBS overnight at 4 °C using: anti-CSF1 (Abcam, USA, SP211, monoclonal antibody, 1:250), anti-TWIST1 (Abcam 10E4E6 monoclonal antibody, 1:100), and anti-CD-68 (Invitrogen, USA, FA-11, monoclonal antibody, 1:200). Sections were incubated with secondary antibodies (Advanced ^TM^ HRP Link, DakoCytomation, Denmark) for half-hour followed by the polymer detection system (Advanced ^TM^ HRP Link, DakoCytomation) for half-hour at room temperature. Reactions were developed using a solution of 0.6 mg/mL of DAB (Sigma, St Louis, MO, USA) and 0.01% H_2_O_2_ and then counter-stained with hematoxylin. Positive controls were included in all reactions in accordance with manufacturer’s recommendations. Negative control consisted in omitting the primary antibody and replacing the primary antibody by normal serum. IHC reactions were replicated on distinct TMA slides to represent different tissues levels in the same lesion. The second slide was 25–30 sections deeper than the first slide, resulting in a minimum of 300 μm distance between sections representing 4-fold redundancy with different cell populations for each tissue.

Two independent certified pathologists conducted the IHC analysis blindly to the clinical data. Cores were scanned in 10× power field to settle on the foremost to marked area predominant in a minimum of 10% of the neoplasia [21]. IHC reaction was considered as positive if of a clearly visible dark brown precipitation occurred. IHC analysis considered the percentage and intensity of staining as: 0 (no detectable reaction or little staining in < 10% of cells), 1 (weak but positive IHC expression in > 10% of cells) and 2 (strong positivity in > 10% of cells) [31,46,49]. Samples were categorized into two groups: 0 (negative) and 1 + 2 (positive cases) for statistical propose.

### 4.7. Statistical Analysis

Statistical analyses of associations between variables were performed by the Fisher’s exact test (with significance set for *p* < 0.05) and for continuous variables the non-parametric Mann–Whitney *u* test. The overall survival was defined as the interval between the beginning of treatment (surgery) and the date of death or the last information for censored observations. Survival probabilities were analyzed by the Kaplan–Meier method and Cox regression models. The log-rank test was applied to assess the significance of differences among actuarial survival curves with a 95% confidence interval. A multivariate Cox proportional hazard models was performed to examine the impact of different predictors on survival. All analyses were performed using the statistical software package STATA-13 (STATA Corporation, College Station, TX, USA) as we previously described [31,46].

### 4.8. Cell Culture and Co-Culture

Oral cancer cell lines SCC9 and SCC25 (ATCC, Manassas, VA, USA) and macrophages RAW264.7 (ATCC) were cultured in DMEM medium (Invitrogen, Carlsbad, CA, USA) supplemented with 10% FBS (Mediatech Inc., Herndon, VA, USA), 400 ng/mL hydrocortisone and 100 µg/mL gentamycin and kanamycin. OSCC1.2/RBT3 cell line was established by our group from a patient with a highly metastatic oral cancer and maintained in culture as previously described [50,51]. In addition, normal epithelial cells (NOE) were isolated from human tongue was maintained in cell culture with KSF serum-free medium supplemented with 5 µg/mL of bovine pituitary extract (Gibco/Invitrogen Life Technology, Carlsbad, CA, USA) as we previously described [52]. In inducing experiments, RAW264.7 cells were incubated with the supernatant from OSCC cells (diluted 1:5) for 48 h. Co-cultivation of macrophages and OSCC cells was performed in 24-wells Boyden chambers (Corning, cat. no. 3413, Tewksbury, MA, USA). Macrophages were seeded on the 0.4 µM inserts, which are permeable to supernatants but not to cellular components. OSCC cells were seed in the lower chambers and grown for 48 h [53]. Cell lines were routinely treated with MycoZAP (Lonza, NJ, USA) and tested for mycoplasma contamination.

### 4.9. Cell Migration and Invasion Assays

Invasion was measured by evaluating the migratory cell rate through a polycarbonate membrane (8-μm pore diameter) coated with BD Matrigel Matrix (BD Biosciences, Bedford, MA, USA) in a modified Boyden chamber (Corning, cat. no. 3422) as previously described [31,46,54]. Collective cell migration was evaluated using qualitative wound-healing assay [31,46]. Each experiment was performed three times and results are expressed as mean ± SD. Statistical analysis was done using the Student’s *t* test.

### 4.10. ELISA Assay for CSF1

Cells were plated at 2 × 10^5^ cells/well into 6-well-plates 48 h before the collection of culture supernatants. The concentrations of CSF1 in the cell culture supernatants were determined using MCSF immunoassay (Quantikine Human M-CSF ELISA Kit, R&D Systems Europe Ltd., Abingdon, UK) according to the manufacturer’s protocols.

### 4.11. siRNA Expression

*TWIST1* target sequences (GenBank, NM_000474) were: 5′-UUGAGGGUCUGAAUCUUGCUCAGCU-3′ and 5′-AGCUGAGCAAGAUUCAGACCCUCAA-3′. Transfections were carried out as previously described [31,46].

### 4.12. CRISPR-Cas9 for the Feneration of TWIST1 Knockout Cells

Two target guide sequences for human (*hTWISTFWD-2*: CACCGCCGCCGAGCGGCAAGCGCGG, *hTWISTREV-2*: AAACCCGCGCTTGCCGCTCGGCGGC; *hTWISTFWD-3*: CACCGGCAAGCGCGGGGGACGCAAG; *hTWISTREV-3*: AAACCTTGCGTCCCCCGCGCTTGCC) and for mouse (*mTwistFWD-2*: CACCGCTGTCGTCGGCCGGAGAGAC; *mTwistREV-2*: AAACGTCTCTCCGGCCGACGACAGC; *mTwistFWD-3*: CACCGACGCAGCAGTCGGCGCAGCG; *mTwistREV-3*: AAACCGCTGCGCCGACTGCTGCGTC) were cloned into lentiCRISPRv2 vector. OSCC1.2 (RBT3, human cell line) and AT84 (mouse cell line) were then transduced by lentiviral particles. The wide-type control was only the lentiCRISPR vector. The transduced cells were selected with puromycin at 1 µg/mL. *TWIST1* knockout clones were confirmed by qRT-PCR and immunoblotting.

### 4.13. Immunoblotting Analysis

Protein extracts were used for immunoblotting assays as we previously described [31,46,54]. Blots were detected using the antibodies for anti-CSF1 (1:1000; Abcam, USA), anti-TWIST1 (1:500; Abnova, Taipei, TW), anti-GAPDH (1:10,000; Cedarlane, Hornby, UK). Secondary antibodies and an enhanced chemiluminescence detection system (Bio-Rad, CA, USA) were used for detecting the Western blot signals. Detailed information about the western blot can be found at Appendix A.

### 4.14. Animal Model

In vivo experiments were carried out in accordance Canadian guidelines (institutional and Federal) after being approved by Animal Care Committee (Protocol # 5018—McGill University) (Research Animal Policy|Procurement Services—McGill University—https://www.mcgill.ca/procurement/regulation/policies/commoditypolicy/animal). 10 × 10^5^ cells were injected orthotopically in the tongue of immunocompetent C3H mice to address tumor microenvironment as we described earlier [31]. Nude mice (20–25 g) were used to study the highly metastatic human cell line (OSCC1.2-RBT3). Mice were sacrificed when the tumor reached 100 mm^3^, and their tumors were dissected, measured, and weighted using a precision balance. The invasive phenotype was evaluated macroscopically and by histological examination (H&E). Results were expressed as the mean ±SD (*n* = 8 per condition) and statistical analysis was done using the Student’s *t* test.

### 4.15. Subcutaneous Matrigel Plug Assay for In Vivo Evaluation of Macrophage Infiltration

To directly analyze macrophage infiltration in immunocompetent mouse model, AT84 cells and AT84 *TWIST1* knockout were mixed with a solution of Matrigel Matrix (BD Biosciences, USA) and implanted subcutaneously in male C3H mice. When this solution reaches body temperature, it jellifies and forms a plug containing the AT84 cells. After 7–15 days, animals were sacrificed, Matrigel plugs were removed, and the characteristics and percentage of macrophage infiltrating cells were analyzed. Images from the whole-mount preparations of the Matrigel specimens were submitted to IHC analysis for TWIST1 (Abcam, 1:100), and CSF1 (Invitrogen, 1:200) proteins.

### 4.16. Macrophage Characterization

RNA isolated from RAW264.7 cell line was reverse transcribed, and qRT-PCR was performed as described above. Gene-specific primers for mouse macrophage characterization by qRT-PCR were: Rn18s (5′-GTAACCCGTTGAACCCCATT-3′; 5′-CCATCCAATCGGTAGTAGCG-3′), Gapdh (5′-TGAAGGTCGGTGTGAACGG-3′; 5′-CGTGAGTGGAGTCATACTGGAA-3), and Actb (5′-CGGTTCCGATGCCCTGAGGCTCTT-3′; 5′-CGTCACACTTCATGATGGATTTGA-3) as internal control; and for macrophage polarization were considered: Nos2 (iNOS, 5′-CCAAGCCCTCACCTACTTCC-3′; 5′-CTCTGAGGGCTGACACAAGG-3′), Tnf (TNF-A, 5′-TCTCATGCACCATCAAGGACT-3′; 5′-TGACCACTCTCCTGCAGAACT-3′), Il6 (IL-6 5′-TTCCATCCAGTTGCCTTCTT-3′; 5′-CAGAATTGCCATTGCACAAC-3′), Il12 (IL-12 5′-GGAAGCACGGCAGCAGAATA-3′; 5′-AACTTGAGGGAGAAGTAGGAATGG-3′), Chil3 (Ym1 5′-CACCATGGCCAAGCTCATTCTTGT -3′; 5′-TATTGGCCTGTCCTTAGCCCAACT -3′), Retnla (Fizz1 5′-ACTGCCTGTGCTTACTCGACT-3′; 5′-AAAGCTGGGTTCACCTCTTCA-3′), Arg1 (5′-AGATTATCGGAGCGCCTTTCT-3′; 5′-TGCTGCAGGGCCTTTCTCT-3′), Il10 (IL-10, 5′-GGTTGCCAAGCCTTATCGGA-3′; 5′-ACCTGCTCCACTGCCTTGCT-3′), Nr3c2 (MR 5′-CCACAGCATTGAGGAGTTTG-3′; 5′-ACAGCTCATCATTTGGCTCA-3′). Relative gene expression analysis was done using Pfaffl model [20].

## 5. Conclusions

This study provides insight into the crosstalk between TWIST1 and CSF1 in metastatic OSCC and supports TWIST1-mediated macrophage activation to promote tumor invasion. Furthermore, the results show the potential of targeting macrophage signaling to manage advanced OSCC, such as using small molecule modulators of macrophage signaling or anti-MIF (migration inhibitory factor; e.g., BAX69 or Imalumab). Some of these agents are currently undergoing clinical trials.

## Figures and Tables

**Figure 1 cancers-13-00153-f001:**
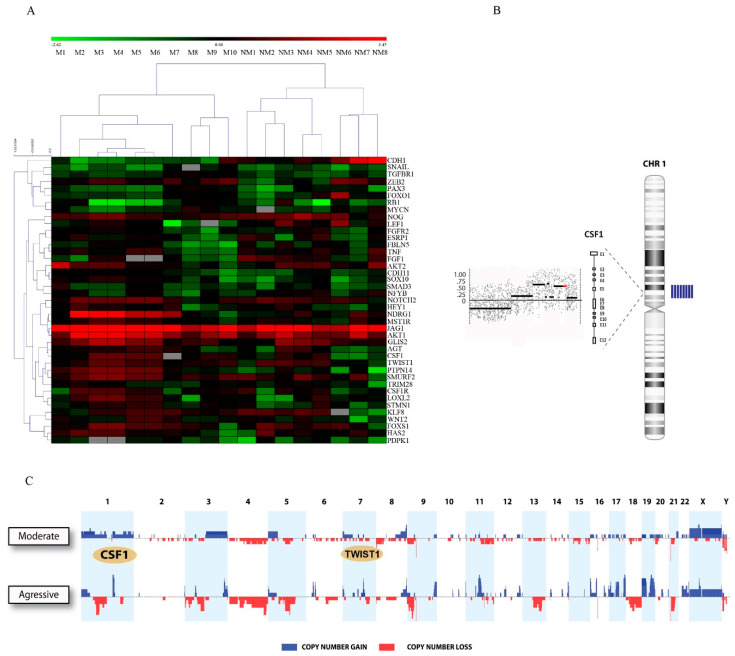
*TWIST1* and *CSF1* were clustered within the gained/amplified chromosomal regions in patients with highly metastatic oral cancer. (**A**) OSCC were sorted using hierarchical unsupervised clustering based on genomic data of EMT genes according to clinical information (M: metastatic vs. NM: non-metastatic OSCC). (**B**) Representation of the chromosome 1 showing the DNA amplification for csf1 sequence in patients with metastatic disease (blue bars in the right of the image). (**C**) The frequency plot identified a large number of chromosomal imbalances in more than 90% of aggressive/metastatic OSCC compared with non-metastatic tumors. The highest scoring *locus* identified in this analysis were 1p13.3 and 7p13 corresponding to *CSF1* and *TWIST1*.

**Figure 2 cancers-13-00153-f002:**
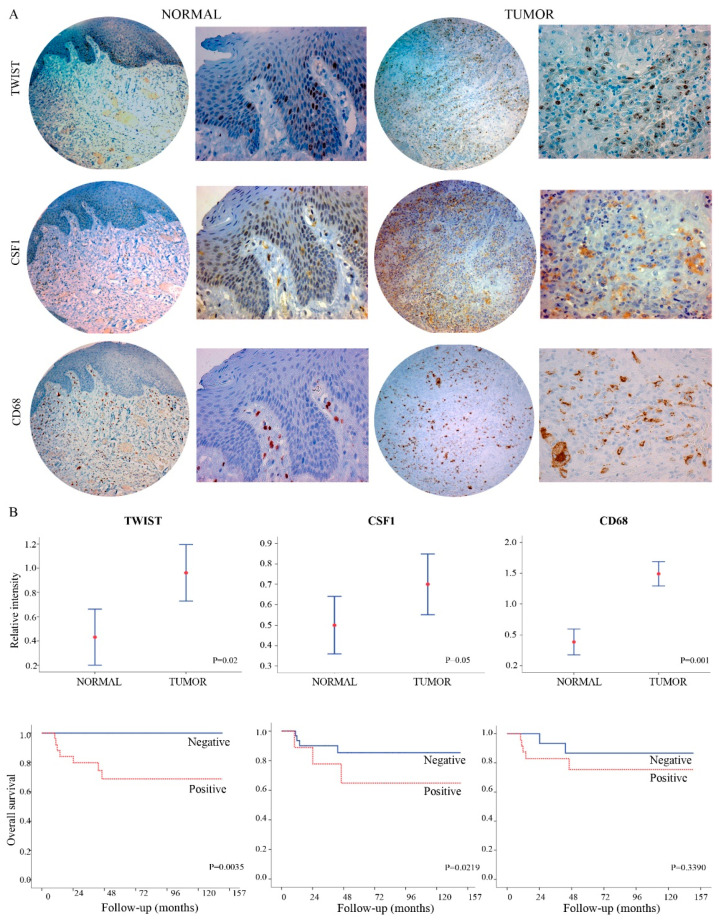
TWIST1 and CSF1 can predict oral cancer progression to metastasis and poor outcomes. (**A**) Immunohistochemistry images of TWIST1, CSF1 and CD68 proteins expression in normal (left side) and oral cancer samples (right side). A negative or weak staining was observed in morphologically normal tissues while a strong nuclear staining (for TWIST1) and cytoplasmic staining (for CSF1 and CD68) was detected in OSCC samples. Magnification of 50× (circle in the left) followed by enlarged area of the image with 200× (square in the right). (**B**) TWIST1, CSF1 and CD68 proteins were differentially expressed in non-metastatic and metastatic OSCC patients as well as in non-cancer tissues. Confidence intervals (CI: 95%) show relative percentage and IHC intensity value. Y-axis represents numerical values corresponding to the percentage and intensity of expression. The bottom graphs represent the Kaplan-Meier overall survival analysis showing a significant positive correlation involving survival rate for TWIST1 and CSF1 expression (log-rank test, *p* = 0.0035, *p* = 0.0219, respectively) but not CD68 (log-rank test, *p* = 0.3390).

**Figure 3 cancers-13-00153-f003:**
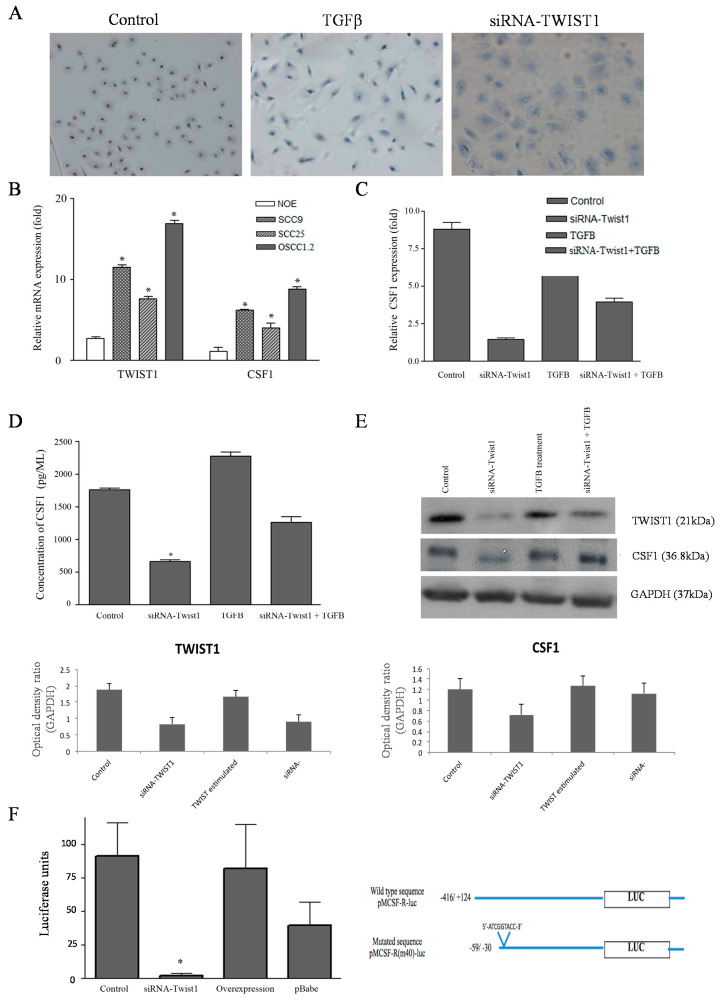
*TWIST1* can regulate *CSF1* expression in oral cancer. (**A**) Morphological aspects of oral normal epithelial cell lines (NOE) after stimulation with TGFβ showing mesenchymal phenotype. The same cells after knockdown *TWIST1* but under stimulation of TGFβ show mixed aspects of epithelial and mesenchymal phenotype. (**B**) Transcript levels of *TWIST1* and *CSF1* estimated by qRT-PCR was increased in a highly metastatic oral cancer cell line (OSCC1.2/RBT3), as well as in the invasive SCC9 cells in comparison to the less-invasive OSCC cell line (SCC25) and normal oral epithelial (NOE) cells. *GAPDH* was used to normalize the gene expression data. Y-axis corresponds to the relative quantification of the mRNA levels and X-axis represents the genes. (**C**) Significant downregulation of *CSF1* was achieved after *TWIST1* gene silencing. *GAPDH* was used as an internal control. Y-axis corresponds to the relative quantification of transcript levels; X-axis represents control OSCC cells, cells expressing siRNA and cells knockdown *TWIST1* and treated with TGFβ. (**D**) Concentration of GMCSF (pg/mL; 2 × 10^5^ cells) measured by ELISA (enzyme-linked immunosorbent assay) in supernatants from OSCC cell culture. Data are represented as mean ± SD (standard deviation) from four separate experiments. Cells downexpressing *TWIST1* shows statistically significant reduction of secreted CSF1. (**E**) Immunoblotting showing efficient down-regulation of TWIST1 and CSF1 (>90%) protein levels compared to control group. GAPDH was used as loading control. (**F**) Transient transfection of pMCSF-R(m40)-luc (5′-ATCGGTACC-3′) into OSCC cells with or without TGFβ treatment revealed that although the mutated constructs displayed constitutive baseline activity, which was significantly increased after TGFβ treatment. (*) Asterisks indicate statistically significant differences in relation to the OSCC untreated cells, where *p*  <  0.05 by ANOVA.

**Figure 4 cancers-13-00153-f004:**
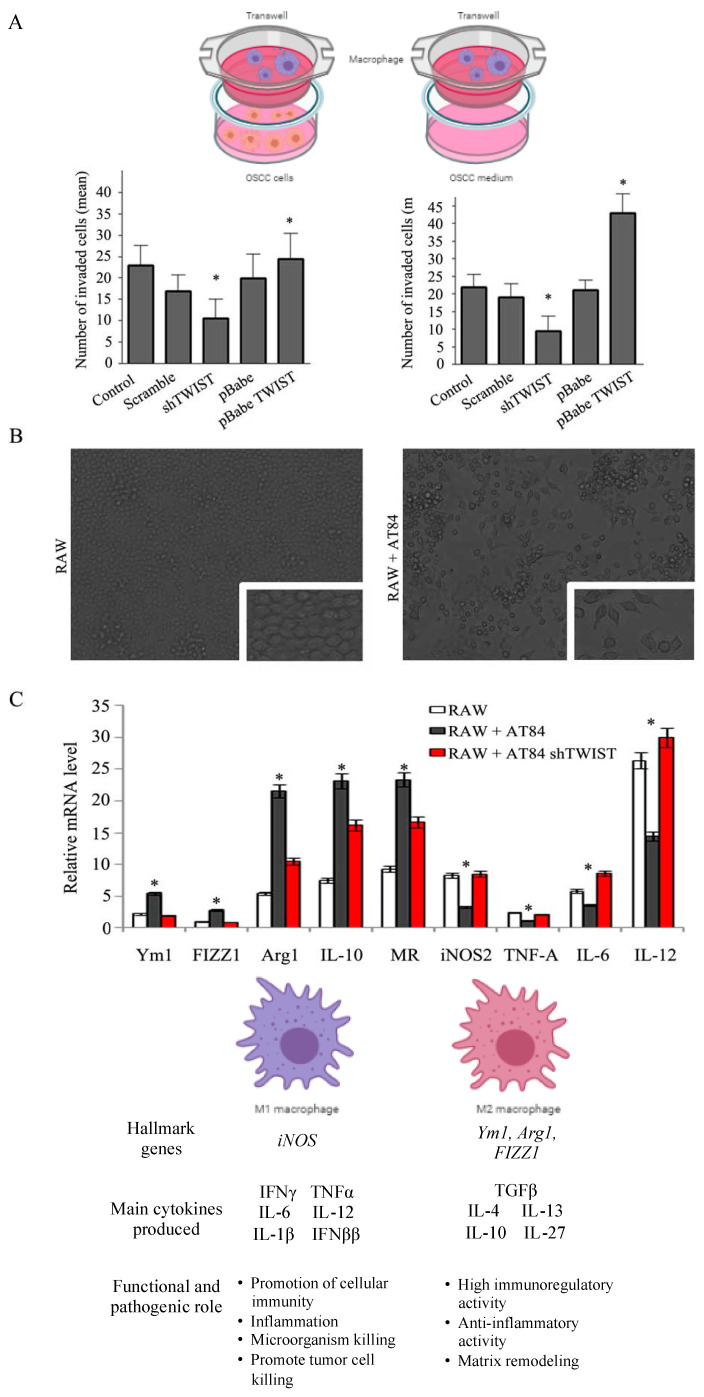
TWIST1 can regulate macrophage chemotaxis and polarization in metastatic oral cancer. (**A**) TWIST1 inhibition leads to decreased macrophage migration. Using a model of migration in co-culture assay, RAW264.7 cells were exposed to AT84, AT84 scramble, AT84 shTWIST1, AT84 pBabe and AT84 overexpressing TWIST1 for 24 h (left side) and their conditioned medium (right side). OSCC cells and medium were used as the chemo-attractant for in vitro cell migration assay. RAW264.7 cells were seeded on top. After 18 h, the migrated cells were stained with hematoxylin for nuclear staining and counted. Error bars represent ± SD of five independent experiments: * *p* < 0.005. (**B**) TWIST1 induced macrophage polarization. Magnification of 10× (big square) followed by enlarged area of the image with 200× (small square in the corner). (**C**) Clear down-expression of classical activated M1 macrophage genes (Nos2/iNOS, Tnf/TNF-A, Il6/IL-6, Il12/IL-12) followed by the increased mRNA expression of alternatively polarized macrophage genes (Chil3/Ym1, Retnla/Fizz1, Arg1, Il10/ IL-10, Nr3c2/ MR). Data are expressed as fold change relative. Bar graph represents the mean ± SD of three independent experiments repeated in triplicates; * *p* < 0.005.

**Figure 5 cancers-13-00153-f005:**
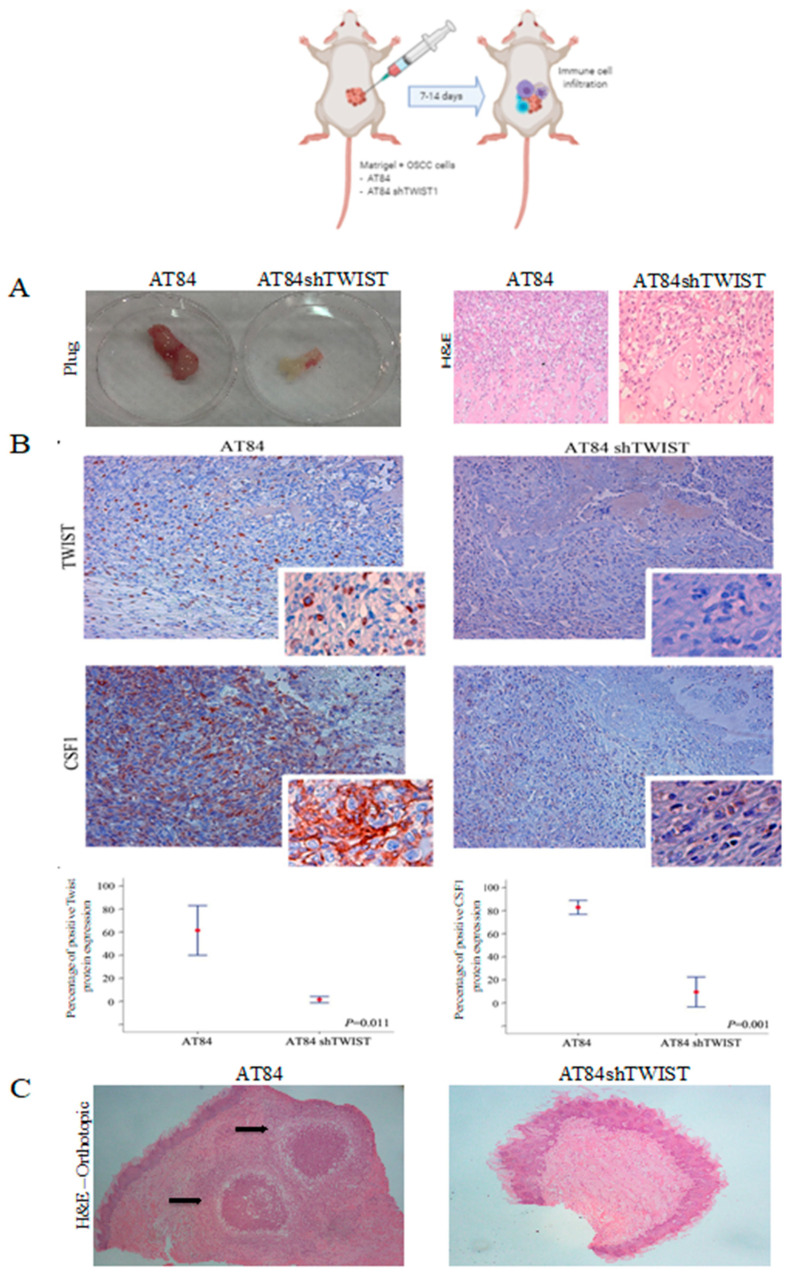
In vivo Matrigel plug assay and orthotopic model (**A**) Plugs obtained from mice 14 days after inoculation with Matrigel containing vehicle (control), AT84 cells and AT84 knockout *TWIST1* were submitted to H&E. Original magnification: 20×. (**B**) Immunohistochemistry images for TWIST1 and CSF1 proteins in AT84 control (left) and AT84 sh*TWIST1* samples (right). Negative staining for TWIST1 and a week expression for CSF1 was observed in the tissues where AT84 sh*TWIST1* was implanted; while a strong nuclear staining (for TWIST1) and cytoplasmic expression (for CSF1) was detected in the plug with AT84 cells. Original magnification: 20× (large rectangle) 200× (small rectangle). (**C**) H&E showing *TWIST1* down-regulation inhibited tumor growth when AT84 cells were implanted either orthotopically into the tongues of mice and no clinical lesion could be observed (compared with AT84 control). Original magnification: 10×.

**Figure 6 cancers-13-00153-f006:**
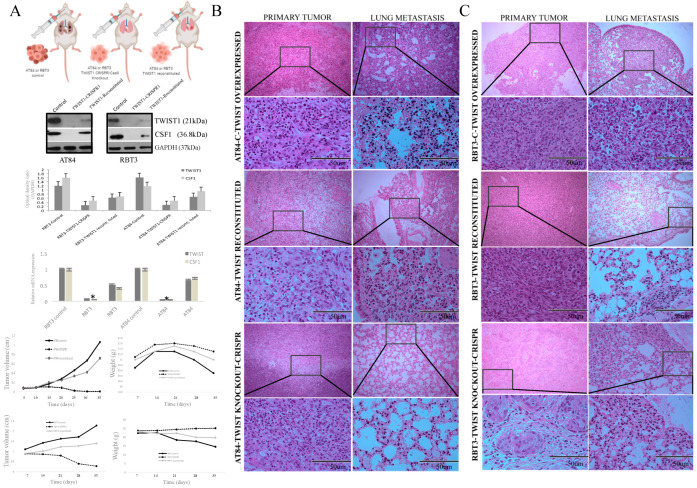
In vivo experiments using murine model (AT84) and a highly metastatic human OSCC (RBT3). (**A**) Immunoblotting and gene expression analysis showing *TWIST1* knockout by CRISPR-CAS9 with significant downregulation of *CSF1* in AT84 and RBT3. The protein and gene expression of both (*TWIST1* and *CSF1*) are present after *TWIST1* be reconstituted in the *TWIST1* knockout tumor cells. GAPDH was used as loading control. Tumor volume and weights are illustrated (*n* = 8) from animals implanted with AT84 and RBT3 (control, *TWIST1* knockout by CRISPR-CAS9, and *TWIST1* reconstituted cells) implanted orthotopically in the tongue. * *p* < 0.005. Representative H&E images using AT84 (**B**) and RBT3 (**C**) cells in the three conditions (control, *TWIST1* knockout by CRISPR-CAS9, and *TWIST1* reconstituted cells). In both experiments, the animas implanted with modified cells (*TWIST1* knockout by CRISPR-CAS9) showed significant inhibition of distant metastasis, while *TWIST1* reconstituted cells increased the metastatic potential. Original magnification: 50× (top image) and 200× (bottom image).

**Figure 7 cancers-13-00153-f007:**
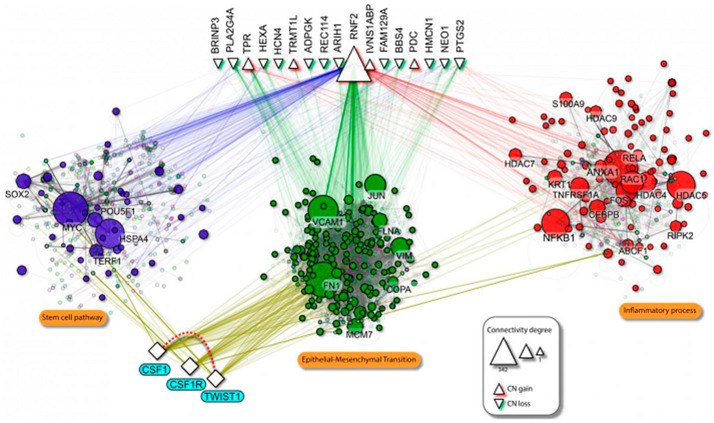
Scheme of the protein-protein interaction (PPI) involving *TWIST1* and *CSF1* network. Solid purple edges represent the interactions between enriched genes in each assessed biological process and the neighborhood interactions. Proteins enrichment was involved in critical hallmarks of cancer progression such as cancer stem cell signaling (purple circle), epithelial-mesenchymal transition (EMT) (green circle), and inflammatory process (blue circle) were detected (*p* < 0.01).

## Data Availability

The data presented in this study are available in Appendix A.

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
