# Peer review of "Co-Overexpression of TWIST1-CSF1 Is a Common Event in Metastatic Oral Cancer and Drives Biologically Aggressive Phenotype"

_cancers, 2021, doi:10.3390/cancers13010153_

Round 1

Reviewer 1 Report

Although the manuscript submitted for evaluation is interesting and original, and numerous several experiments were performed, certain not very robust aspects should be improved before its consideration for publication. Therefore, I cannot recommend its acceptance in its current state:

INTRODUCTION

- Introduction is too short and could be better elaborated.  A more explicit and real paragraph showing oral cancer epidemiology (minimally no. cases-deaths/year) should be written.

- References should also be carefully revised and updated. For example, “4. Huang S.H., O'Sullivan B. Oral cancer: Current role of radiotherapy and chemotherapy. Med 531 Oral Patol Oral Cir Bucal 2013;1 e233-240. 532 5. Scully C., Bagan J.V. Recent advances in oral oncology; squamous cell carcinoma imaging, 533 treatment, prognostication and treatment outcomes. Oral Oncol 2009; 45: e25-30.” are not appropiate nor up-to-date references. GLOBOCAN last report is suggested.

- I also miss a better elaborated objectives paragraph. More detailed data on population under study, and outcomes evaluated and methodology setting should be reported.

 METHODS

- Consensus reporting and/or methodological guidelines were not reported for any analysis (e.g., STROBE for the cohort or ARRIVE for the animal experiments).

- The genomic experiments were performed in a very low sample size (10 metastatic vs 10 non-metastatic tissues). I strongly doubt that solid conclusions can be reached with such a sample size, very probably underpowered. Figure 1 depicts the results from the copy number alterations analysis (figure is pixelated with not very high resolution, making complex its evaluation), and only 8 patients are represented in the hierarchical clustering plot. The data from the two remaining patients is missing.

- The methodology used for the immunohistochemical analysis could be better described. “anti-CSF1 394 (Abcam, Cambridge, 1:250), anti-TWIST1 (Abcam, 1:100), and anti-CD-68 (Invitrogen, 1:200)”. Authors should also report the clones used, not only commercial brands, and their nature (mono/polyclonal). The scoring system could also be improved, allowing the repeatability of methods. How intensity and cell count were integrated to obtain a final score?

- The origin of the validation cohort was not reported (from Brazil or Canada? Brazil is expected to a correct validation, but It should be confirmed by the authors).

- One of the main methodological problems is the consideration of “Matched morphologically normal specimens from the surgical margins” as healthy control groups. Accordingly, the methodology of this work is seriously biased (low internal validity), since the presence of genetically altered premalignant fields is well established in patients with head and neck cancer. The field cancerization theory is well known and accepted in this anatomical area, where the oral mucosa apparently clinically healthy harbors early oncogenic molecular alterations, with very important prognostic implications. For this reason, the adjacent non-tumor tissue of these patients should not be used as a control group, and It should not be considered as a "normal" or healthy sample. Therefore, the conclusions of the present work are probably biased due to not having selected an adequate control group.

- In relation with lab in vitro experiments, only three cell lines were cultured. This is also an important limitation, due to the low repeatability of these experiments. Furthermore, it is not clear whether positive/negative controls were used in these experiments. Finally, neither in vitro nor in vivo animal experiments were blinded (only the histopathologic analysis of human tissues), a good practice seldom performed in these types of analyses.

- “Statistical analyses were performed using the software package STATA-13 (STATA Corporation, College Station) as we previous described”. Statistical section should also be comprehensively described, minimally, all the variables to be analyzed should be listed, jointly with the tests applied.

RESULTS

Although statistical methods were not reported, the analysis could be clearly improved. For survival variables were only reported Kaplan-meier curves with p-values (derived from log-rank method). As time-to-event variables, hazard ratios with their corresponding confidence intervals should be estimated and, if possible, adjusted at multivariable level.

The last commentary also applies to clinico-pathological variables, only presented by “crude” data in supplemental tables. For this analysis, clinical stage and histological grade (two relevant variables, always present for the individual evaluation of patients) were not included in the analysis.

Only data for CSF1 were presented in tables. The data derived from the analysis of the rest of biomarkers under study (e.g., Twist) should also be presented to avoid a potential selective reporting bias.

DISCUSSION

- A paragraph with limitations also lacks. High impact journals consider it mandatory. Please, be autocritical and really discuss the study limitations.

- A paragraph including recommendations for future studies would also be highly recommended.

Although I understand that the suggestions made in my review are complicated to carry out in their enterity, it is imperative for my final approval of this manuscript to make all the suggested changes.

Author Response

Reviewer #1:

Comment: - Introduction is too short and could be better elaborated. A more explicit and real paragraph showing oral cancer epidemiology (minimally no. cases-deaths/year) should be written.

Response: As suggested we have expanded the introduction to address oral cancer epidemiology (page 2, lines 46-53) as follow:

“Oral cancer is the most common subtype of malignant tumors of the head and neck, which represents the 6th most frequent cancer worldwide with approximately half million cases and 300,000 deaths annually (4-6). Advances in surgical procedures and therapeutic approaches for oral squamous cell carcinoma (OSCC) have led to a substantial improvement in survival rates but the overall 5-year relative survival is lower than 50% and it remains inferior compared to highly frequent cancers such as breast, prostate and lungs cancers (7,8). The high incidence of tumor relapse and distant metastasis are the main contributors of OSCC-related mortality (9)”.

Comment: - References should also be carefully revised and updated. For example, “4. Huang S.H., O'Sullivan B. Oral cancer: Current role of radiotherapy and chemotherapy. Med 531 Oral Patol Oral Cir Bucal 2013;1 e233-240. 532 5. Scully C., Bagan J.V. Recent advances in oral oncology; squamous cell carcinoma imaging, 533 treatment, prognostication and treatment outcomes. Oral Oncol 2009; 45: e25-30.” are not appropiate nor up-to-date references. GLOBOCAN last report is suggested.

Response: We revised and updated our list of references (pages 18-21), and GLOBOCAN last report was included as suggested (page 18, lines 563-568, References 3 and 4).

Comment:  I also miss a better elaborated objectives paragraph. More detailed data on population under study, and outcomes evaluated and methodology setting should be reported.

Response: We re-wrote the last paragraph of the introduction to provide more details about the study population and methodology overview (page 2; lines 74-81) as follow:

“In this study, we conducted a comprehensive genome-wide screening, and it was identified a common co-amplification of TWIST1-CSF1 in cancer tissues using a unique cohort of patients with highly metastatic OSCC compared with non-metastatic ones. The clinicopathological impact of this co-overexpression was validated in a large cohort of OSCC patients. We further investigated the mechanistic implication of TWIST1-CSF1 signaling to macrophage chemotaxis and polarization by showing a role of TWIST1 in the remodeling of OSCC tumor microenvironment via CSF1 regulation; this enhanced OSCC progression and metastasis competence in vitro and in vivo.”

Comment: - Consensus reporting and/or methodological guidelines were not reported for any analysis (e.g., STROBE for the cohort or ARRIVE for the animal experiments).

Response: We cited the statement for observational studies (The Strengthening the Reporting of Observational Studies in Epidemiology - STROBE) and ARRIVE as specific guidelines (Animals in Research: Reporting in vivo - ARRIVE). STROBE was reported in the last paragraph of “Study population” section (Material and Methods page 14; lines 374-376) as follow:

“Strengthening the reporting of observational studies (STROBE Statement) was used to ensure appropriate methodological quality (http://www.strobe-statement.org/)”.

ARRIVE specific guidelines is under the reference reported in the first sentence of animal model description (Research Animal Policy | Procurement Services - McGill University - https://www.mcgill.ca/procurement/regulation/policies/commoditypolicy/animal) which contain the requirements and criteria for experimental design used in this study, including sample size calculation, inclusion and exclusion criteria, blinded assessment of outcome that is the minimum requested criteria by Canadian Guidelines (Institutional and Federal) (Materials and methods section; page 17; lines 495-498) as follow:

“In vivo experiments were carried out in accordance Canadian guidelines (institutional and Federal) after be approved by Animal Care Committee (McGill University) (Research Animal Policy | Procurement Services - McGill University - https://www.mcgill.ca/procurement/regulation/policies/commoditypolicy/animal).”

Comment: - The genomic experiments were performed in a very low sample size (10 metastatic vs 10 non-metastatic tissues). I strongly doubt that solid conclusions can be reached with such a sample size, very probably underpowered. Figure 1 depicts the results from the copy number alterations analysis (figure is pixelated with not very high resolution, making complex its evaluation).

Response: We agree that larger size cohorts would be suitable. However, it is hard to get larger cohorts with frozen samples from patients with metastatic oral cancer with a longer follow-up. We still believe the high quality of our matched cohorts with years follow-up are unique since cases were selected based on strict clinicopathological criteria to come up with a homogenous cohort as possible: only one tumor location (tongue), same treatment, same histological grade and microscopic invasion patterns and similar patients’ outcomes (metastatic versus non-metastatic cases). Besides, all these samples were submitted to laser capture microdissection to guarantee homogenous cell population with only epithelial cells to be analysed (there was no necrosis, no inflammatory infiltration, and any other potential contaminant from stroma cells). Furthermore, results from these cohorts were validated in a much large cohort of oral cancer patients in addition to supporting data from preclinical and animal models showing how the data is strong.

Comment: - The methodology used for the immunohistochemical analysis could be better described. “anti-CSF1 394 (Abcam, Cambridge, 1:250), anti-TWIST1 (Abcam, 1:100), and anti-CD-68 (Invitrogen, 1:200)”. Authors should also report the clones used, not only commercial brands, and their nature (mono/polyclonal). The scoring system could also be improved, allowing the repeatability of methods. How intensity and cell count were integrated to obtain a final score?

Response: We have improved the methodology description as recommended. The intensity and cell count was grouped and for statistical analysis and we had two groups: negative (no expression or little staining in < 10% of cells) and 1+2 (weak/intermediate or strong protein expression in more than 10% of the tissue). These criteria were clarified in the Materials and Methods (pages 15 and 16; lines 413-433) as follow:

“4.6. Immunohistochemistry (IHC) and statistical analysis

Immunohistochemistry reaction was carried out on the TMA as we described (31, 46). In brief, the slides were incubated with primary antibodies diluted in PBS overnight at 4oC using: anti-CSF1 (Abcam SP211, monoclonal antibody, 1:250), anti-TWIST1 (Abcam 10E4E6 monoclonal antibody, 1:100), and anti-CD-68 (Invitrogen FA-11, monoclonal antibody, 1:200). Sections were incubated with secondary antibodies (Advanced TM HRP Link, DakoCytomation, Denmark) for half-hour followed by the polymer detection system (Advanced TM HRP Link, DakoCytomation) for half-hour at room temperature. Reactions were developed using a solution of 0.6mg/mL of DAB (Sigma, St Louis) and 0.01% H2O2 and then counter-stained with hematoxylin. Positive controls were included in all reactions in accordance with manufacturer´s recommendations. Negative control consisted in omitting the primary antibody and replacing the primary antibody by normal serum. IHC reactions were replicated on distinct TMA slides to represent different tissues levels in the same lesion. The second slide was 25-30 sections deeper than the first slide, resulting in a minimum of 300um distance between sections representing 4-fold redundancy with different cell populations for each tissue.

Two independent certified pathologists conducted the IHC analysis blindly to the clinical data. Cores were scanned in 10X power field to settle on the foremost to marked area predominant in a minimum of 10% of the neoplasia (21). IHC reaction was considered as positive if of a clearly visible dark brown precipitation occurred. IHC analysis considered the percentage and intensity of staining as: 0 (no detectable reaction or little staining in < 10% of cells), 1 (weak but positive IHC expression in > 10% of cells) and 2 (strong positivity in > 10% of cells) (31, 46, 49). Samples were categorized into two groups: 0 (negative) and 1+2 (positive cases) for statistical propose.”

Comment: - The origin of the validation cohort was not reported (from Brazil or Canada? Brazil is expected to a correct validation, but It should be confirmed by the authors).

Response: Initially we conducted technical validation from samples obtained from Brazil and then biological validation was conducted in an independent cohort from Canada. We clarified this issue in the section Materials and Methods (page 14, lines 359-363) as follow:

“Technical validation (cohort from Brazil) and validation of the biological process (independent cohort from Canada) were done in 141 paraffin-embedded oral cancer specimens from patients who had tumor relapse (n=44; 31.2%) or distant metastasis and patients with good outcomes (n=97; 68.8%) were evaluated by immunohistochemistry (IHC) using tissue microarray (TMA).”

Comment: - One of the main methodological problems is the consideration of “Matched morphologically normal specimens from the surgical margins” as healthy control groups. Accordingly, the methodology of this work is seriously biased (low internal validity), since the presence of genetically altered premalignant fields is well established in patients with head and neck cancer. The field cancerization theory is well known and accepted in this anatomical area, where the oral mucosa apparently clinically healthy harbors early oncogenic molecular alterations, with very important prognostic implications. For this reason, the adjacent non-tumor tissue of these patients should not be used as a control group, and It should not be considered as a "normal" or healthy sample. Therefore, the conclusions of the present work are probably biased due to not having selected an adequate control group.

Response: We apologize if our methods was not clear. For the genomic analysis, normal tissues were commercially acquired for the experiments (Promega, Madison, WI, USA) (page 15, lines 385-386). Matched morphologically epithelium margins were included in the tissue microarray (TMA) for immunohistochemistry analysis (Figure 2A and Supplemental Material – Figure 1). As explained in Materials and Methods (page 15; line 407-409; References 31 and 46), our TMA were constructed with samples from OSCC (primary tumor), morphological normal samples (margins from the same patients) and lymph nodes (when available). Morphological normal tissues were not considered as healthy control in our study. Even thought the field cancerization theory may exist in oral cavity from patients with OSCC, our results showed clear differences in the protein expression (please see our images and graphs in the Figure 2) between non-tumor tissue versus tumor samples, showing that the protein is strongly overexpressed in tumors going to EMT (metastatic tumors). Our goal was to show that TWIST1-CSF1 overexpression is a common event in metastatic oral cancer (compared with non-metastatic tumors). This was the conclusion and the title of our manuscript. It was not our aim to explore oral carcinogenesis and potential gene signaling and/or protein expression in tissues from non-tumor specimens (this could be a subject for another study). In this sense, there is no bias in our conclusion and we are confident in our results. In addition, we would like to stress that the experimental design follow by the validation (clinical, preclinical and in animal model) support our claims.

Comment: - In relation with lab in vitro experiments, only three cell lines were cultured. This is also an important limitation, due to the low repeatability of these experiments. Furthermore, it is not clear whether positive/negative controls were used in these experiments. Finally, neither in vitro nor in vivo animal experiments were blinded (only the histopathologic analysis of human tissues), a good practice seldom performed in these types of analyses.

Response: We used three human oral cancer cell lines, one mouse oral cancer cell line and a normal epithelium cell line. In vitro experiments (cell migration, proliferation, invasion, ELISA, macrophage characterization) were repeated at least 3 times independently. Appropriate controls were included for each experiment (siRNA, shRNA or CRISPR-Cas9) using empty plasmid or lentiviral vectors. We also conducted experiments with cell stimulation and/or reconstitution of the protein expression. Gene expression analysis were done in triplicates (per reaction) in at least 3 independent experiments (total of replicates = 9X) to ensure consistency in the data and allow proper statistical analysis. For in vivo experiments, experimental groups were blindly coded so the person who did mice inoculation and monitoring (JS) had no knowledge on group assignments.

Comment: - “Statistical analyses were performed using the software package STATA-13 (STATA Corporation, College Station) as we previous described”. Statistical section should also be comprehensively described, minimally, all the variables to be analyzed should be listed, jointly with the tests applied.

Response: We included statistical analysis as suggested in a separate topic (4.7) in the Materials and Methods (page16, lines 435-444) as follow:

“4.7. Statistical analysis

Statistical analyses of associations between variables were performed by the Fisher’s exact test (with significance set for P<0.05) and for continuous variables the non-parametric Mann–Whitney U test. The overall survival was defined as the interval between the beginning of treatment (surgery) and the date of death or the last information for censored observations. Survival probabilities were analyzed by the Kaplan–Meier method and Cox regression models. The log-rank test was applied to assess the significance of differences among actuarial survival curves with a 95% confidence interval. A multivariate Cox proportional hazard models was performed to examine the impact of different predictors on survival. All analyses were performed using the statistical software package STATA-13 (STATA Corporation, College Station) as we previous described (31, 46).”

Comment: - Although statistical methods were not reported, the analysis could be clearly improved. For survival variables were only reported Kaplan-meier curves with p-values (derived from log-rank method). As time-to-event variables, hazard ratios with their corresponding confidence intervals should be estimated and, if possible, adjusted at multivariable level.

Response: The hazard ratios with their corresponding confidence intervals were added as you suggested in the results as follow (page 4; lines 123-128).

“To investigate whether TWIST1, CSF1, and CD68 expression was associated with patients’ outcomes, Kaplan-Meir and cox proportional hazard models wee performed. A worst overall survival probability was experience by patients with TWIST1 [log-rank test, P=0.0035; adjusted HR 7.837 (95% CI: 1.099-50.880; P=0.040] and CSF1 overexpression [log-rank test, P=0.0219; adjusted HR 2.182 (95% CI: 0.993-4.796; P=0.052)] but not CD68 [log-rank test, P=0.3390; adjusted HR 1.004 (95% CI: 0.987-1.021; P=0.628)] (Figure 2C).”

Comment: - The last commentary also applies to clinico-pathological variables, only presented by “crude” data in supplemental tables. For this analysis, clinical stage and histological grade (two relevant variables, always present for the individual evaluation of patients) were not included in the analysis.

Response: Histological grade data was added in the Supplemental Table as suggested. Clinical stage is representing in the table by the variables T stage (grouped as T1 + T2 versus T3 + T4) and nodal status (N0 versus N+). The inclusion criteria included only M0 patients at initial diagnosis (described in Materials and Methods; page 14, lines 370-373).

Comment: - Only data for CSF1 were presented in tables. The data derived from the analysis of the rest of biomarkers under study (e.g., Twist) should also be presented to avoid a potential selective reporting bias.

Response: Twist-1 expression data is now presented in the Supplemental Table 2 as you suggested.

Comment: - A paragraph with limitations also lacks. High impact journals consider it mandatory. Please, be autocritical and really discuss the study limitations.

Response: The paragraph with limitation was added (pages 12 and 13; lines 305-320) in the discussion section as follow:

“Oral cancer cells undergoing to EMT may not only contribute to increase metastatic competence but may become resistant to cytotoxic T-lymphocytes. We also investigated the presence of T cells in epithelial tumor islets (intraepithelial tumor-infiltrating lymphocytes, TIL) by quantified all CD3+ T cells and cytotoxic CD8+ T cells (Supplemental Figure 1). TIL status was differentially expressed comparing normal, OSCC and lymph nodes, but it was not statistically significant for OSCC patients’ outcomes.  However, using the same cohort, TWIST1 and CSF1 were predictive of OSCC progression and poor prognosis. It should be noted that EMT is a reversible trans-differentiation program with inherent plasticity associated with the stemness of cancer cells sharing considerable redundancies such as mediators, factors, signal transducers and these are not induced simultaneously, We identified that the co-overexpression of TWIST1-CSF1 drives biologically aggressive phenotype in a pure epithelial cell population (our samples were microdissected) from patients with oral cancer presenting very similar clinicopathological characteristics and outcomes. However, to determine if patients may or may not respond to immunotherapy, future researchers should be able to measure the degree of tumor cell undergoing to EMT considering inter and intra-tumor heterogeneity associated with the microenvironment, which is heterogeneous as well.”

Comment: - A paragraph including recommendations for future studies would also be highly recommended.

Response: Conclusion included future directions (page 18, lines 533-537) as follow:

“This study provides insight in the crosstalk between TWIST1 and CSF1 in metastatic OSCC and supports TWIST1-mediated macrophage activation to promote tumor invasion. Furthermore, the results show the potential of targeting macrophage signaling to manage advanced OSCC, such as using small molecule modulators of macrophage signaling or anti-MIF (migration inhibitory factor; e.g. BAX69 or Imalumab), some of these agents are currently undergoing clinical trials.”

Thank you for your time to provide us great comments and suggestions.

Reviewer 2 Report

This study aims to investigate the expression of Twist1-Csf1 in Metastatic Oral Cancer. The study suggests that overexpression of TWIST1, epithelial-mesenchymal-transition (EMT), CSF1, tumor-associated macrophages (TAMs) these markers strongly predicted poor patients' survival. In addition, TWIST1 gene overexpression induces the activation of CSF1. These findings provide the insight in the cooperation between TWIST1 transcription factor and CSF1 to promote OSCC invasiveness. The relate research articles still not published. This manuscript content is suitable for publication in Cancers.

Major comment

In figure 3, the authors point that TWIST1 can regulate CSF1 expression by knockdown TWIST1. Significant downregulation of CSF1 was achieved after TWIST1 gene silencing. Whether the expression of TWIST1 was decreased after CSF1 gene silence?

Author Response

Reviewer #2:

Comment: -This study aims to investigate the expression of Twist1-Csf1 in Metastatic Oral Cancer. The study suggests that overexpression of TWIST1, epithelial-mesenchymal-transition (EMT), CSF1, tumor-associated macrophages (TAMs) these markers strongly predicted poor patients' survival. In addition, TWIST1 gene overexpression induces the activation of CSF1. These findings provide the insight in the cooperation between TWIST1 transcription factor and CSF1 to promote OSCC invasiveness. The relate research articles still not published. This manuscript content is suitable for publication in Cancers.

Major comment:  In figure 3, the authors point that TWIST1 can regulate CSF1 expression by knockdown TWIST1. Significant downregulation of CSF1 was achieved after TWIST1 gene silencing. Whether the expression of TWIST1 was decreased after CSF1 gene silence?

Response: It has been reported that CSF1 inhibition may lead to reduced cell invasion and metastasis in breast cancer*. In this regulatory loop, cancer cells secrete CSF1 to recruit and stimulate macrophages. In turn, macrophages secrete specific growth factors, in particular epidermal growth factor (EGF) to stimulate tumor cells to migrate and metastasize. However, the transcription factors and coactivators that regulate CSF1 expression in cancer cells remains poorly defined. In our study, we demonstrate for the first time, that TWIST1 is a critical transcriptional regulator of CSF1. We are considering alternative future studies because even though CSF1 was knockout/knockdown, TWIST1 is still being stimulated by probably other cell population within the tumor stroma at least in part through the EMT process.

* Ding J, Guo C, Hu P, Chen J, Liu Q, Wu X, Cao Y, Wu J. CSF1 is involved in breast cancer progression through inducing monocyte differentiation and homing. Int J Oncol. 2016 Nov;49(5):2064-2074.

Cannarile MA, Weisser M, Jacob W, Jegg AM, Ries CH, Rüttinger D. Colony-stimulating factor 1 receptor (CSF1R) inhibitors in cancer therapy. J Immunother Cancer. 2017 Jul 18;5(1):53.

Thank you for your excellent question.

Round 2

Reviewer 1 Report

There has only been one unresolved question, the small sample size has not been increased. I understand that this could be very difficult matter.